# Comparative analysis of Egypt's malaria elimination strategies and implementation science: Pathways to achieve malaria-free status for other African countries

**Chukwuka Elendu**[1]*, **Dependable C. Amaechi**[2], **Rhoda C. Elendu**[3],
**Sehajmeet Kaur Saggi**[4], **Tochi C. Elendu**[5], **Emmanuel C. Amaechi**[6], **Ijeoma D. Elendu**[5],
**Kanishk Dang**[7], **Opeyemi P. Amosu**[8], **Omoyelemi F. Idowu**[9], **Toluwanimi S. Oseni**[10]

**1** Federal University Teaching Hospital, Owerri, Nigeria, **2** Igbinedion University, Okada, Nigeria, **3** van Horbachevsky Ternopil National Medical University, Ternopil, Ukraine, **4** Belarusian State Medical University, Minsk, Belarus, **5** Imo State University, Owerri, Nigeria, **6** Madonna University, Elele, Nigeria, **7** Nicolae Testemițanu State University of Medicine and Pharmacy, Chişinău, Republic of Moldova, **8** University of Ilorin Teaching Hospital, Ilorin, Nigeria, **9** Babcock University Teaching Hospital, Ilishan-Remo, Nigeria, **10** R-Jolad Hospital, Lagos, Nigeria

* elenduchukwuka@yahoo.com

## Abstract

### Background

Malaria elimination is a critical public health goal, particularly in Africa, where the disease disproportionately affects vulnerable populations. Egypt's success in achieving World Health Organization (WHO) malaria-free certification on October 20, 2024, through an evidence-based malaria elimination program, offers a valuable model for replication. Our study is identified as an implementation study, evaluating the evidence-based interventions deployed, the implementation strategy tested, and its outcomes to provide insights for scaling similar programs across Africa. A RE-AIM-informed lens was applied to better articulate how contextual factors, intervention reach, effectiveness, adoption, implementation, and maintenance influenced observed outcomes.

### Methods

We adopted a retrospective implementation science approach to analyze Egypt's malaria elimination program. The implementation strategy focused on high-priority malaria-endemic regions, targeting vulnerable groups such as children under five and pregnant women. Interventions included vector control measures, such as insecticide-treated nets (ITNs), indoor residual spraying (IRS), antimalarial treatment protocols, and public health education campaigns. Key implementation strategies included workforce training, multilevel stakeholder engagement, integrated data-driven decision-making, and community mobilization. Our study evaluated key implementation outcomes, including fidelity, feasibility, acceptability (measured using a

**Data availability statement:** All data supporting our study's findings were obtained from publicly available sources, including program reports and published literature, and are fully cited and described in the published article; no primary data were collected.

**Funding:** The author(s) received no specific funding for this work.

**Competing interests:** The authors have declared that no competing interests exist.

custom-designed tool assessing stakeholder and community perceptions of the interventions), and adaptability alongside health outcomes such as incidence reduction, treatment coverage, and community awareness. Economic evaluations and process analyses provided additional insights into cost-effectiveness and operational efficiency.

## Results

The implementation strategy targeted malaria-endemic regions, achieving a 92% recruitment rate, focusing on vulnerable groups such as children under five (37%) and pregnant women (12%). 78% of the recipient population adhered to preventive measures like insecticide-treated nets (ITNs). The program reduced malaria incidence by 96% over 15 years and achieved a 94% reduction in Anopheles mosquito density. Active surveillance led to the detection of 98% of cases within 48 hours of symptom onset, while treatment coverage reached 91%. Community awareness of malaria prevention increased to 84% by the program's conclusion. Economic evaluations revealed a cost per disability-adjusted life year (DALY) averted of $24, with an estimated $1.5 billion saved in healthcare costs and productivity losses over 15 years. Sub-group analyses highlighted higher adherence rates in urban areas (89%) compared to rural areas (73%) and significant reductions in malaria-related complications among pregnant women (78%). Implementation outcomes included high fidelity (93%) to planned strategies, high feasibility across urban and rural contexts, and successful adaptability to emerging challenges such as insecticide resistance and funding fluctuations. Routine monitoring systems, continuous feedback loops, and responsive adaptation mechanisms were central to achieving these outcomes. Acceptability scores averaged 87% across stakeholders, reflecting strong alignment with community values, trust in health authorities, and perceived relevance of interventions.

## Conclusion

Egypt's malaria elimination strategies exemplify the effective integration of implementation science into public health programs. Key success factors included community engagement, robust surveillance systems, and cost-effective resource allocation. The intentional use of implementation strategies and documented outcomes demonstrate alignment with the RE-AIM framework, reinforcing the program's relevance for broader application. Future efforts should emphasize tailored interventions, capacity building, transparent assessment of acceptability, and sustainable funding mechanisms to replicate Egypt's success.

## Introduction and background

Malaria, a parasitic disease caused by Plasmodium species and transmitted primarily by Anopheles mosquitoes, remains a significant public health concern across many African countries. Despite global progress in malaria control and elimination,

the African region bears over 90% of the global malaria burden, accounting for the majority of cases and deaths reported annually [1,2]. However, some countries like Egypt have successfully transitioned to malaria-free status, offering valuable insights into effective strategies and policies that can guide similar achievements in other endemic African nations [3]. The success story of Egypt demonstrates the potential of tailored elimination strategies underpinned by robust implementation science to address the epidemiological, environmental, and sociopolitical dimensions of malaria elimination [4,5].

Historically, malaria was widespread in Egypt, particularly in the southern regions along the Nile River and Delta. Early efforts at control began in the early 20th century, focusing on vector reduction through drainage projects and larviciding [6]. The Global Malaria Eradication Programme (GMEP), launched in 1955 by the World Health Organization (WHO), aimed to eradicate malaria through a globally coordinated effort primarily focused on indoor residual spraying with DDT, mass drug administration, and case detection. The program was designed to target regions where malaria transmission was deemed technically feasible to interrupt, particularly in temperate and subtropical zones with seasonal transmission [7]. Although GMEP failed to achieve its global goals, Egypt's integration of these efforts into its health system established the foundation for later elimination successes [8,9]. A combination of historical experience, epidemiological data, and operational research informed Egypt's selection of specific elimination strategies during the 1970s and 1980s. The country conducted localized entomological and epidemiological studies to identify transmission hotspots and dominant vector species. This evidence-based approach allowed for the customization of interventions such as targeted indoor residual spraying, community-based surveillance, and stratified treatment regimens tailored to the local context [10,11]. By 2024, Egypt was certified malaria-free by the World Health Organization (WHO), marking an important milestone in regional disease elimination efforts [12,13]. Given Egypt's success, our study applies implementation science (IS) to retrospectively identify and analyze the key strategies that supported malaria elimination. This approach allows for a systematic understanding of how evidence-based interventions were selected, adapted, and sustained within Egypt's specific context, thereby offering a transferable framework for other malaria-endemic countries aiming for elimination.

## Statement of concrete aims

We focus on three key implementation outcomes through an implementation science lens: fidelity, feasibility, and adaptability. In addition to identifying core strategies, our study explores contextual factors—such as political commitment, health system integration, and community engagement—that acted as barriers or facilitators to implementation. By systematically documenting these factors and outcomes, our study provides a model for malaria-endemic countries seeking to translate evidence into action.

## Materials and methods

Our study employed a retrospective mixed-methods implementation research design to evaluate Egypt's malaria elimination program. The evaluation adhered to the Standards for Reporting Implementation Studies (StaRI) and was conducted between October 23, 2024, and November 30, 2024, following ethical approval granted on October 22, 2024, by the Federal Ministry of Health Nigeria Research Ethics Committee (FMOH-NREC; Approval No. FMOH/ER/2024/0456). All procedures adhered to institutional ethical standards and the principles outlined in the 1964 Helsinki Declaration and its subsequent amendments. Due to the retrospective nature of this study, informed consent was waived by FMOH-NREC, with all patient data anonymized to ensure confidentiality and privacy. This period was selected to ensure the inclusion of comprehensive and relevant data for a comparative analysis of Egypt's malaria elimination strategies and the implementation of science pathways that may guide other African countries in achieving malaria-free status.

To guide the evaluation, we adopted a hybrid framework approach combining the RE-AIM framework (Reach, Effectiveness, Adoption, Implementation, and Maintenance) with Proctor et al.'s taxonomy of implementation outcomes. RE-AIM provided the overarching structure for assessing public health impact, while Proctor's framework informed the operational definitions and measurement of specific implementation outcomes.

## Study location and setting

The evaluation focused on malaria-endemic regions in Egypt, particularly the Nile Delta and southern governorates, which historically reported high malaria transmission. These regions were selected based on historical documentation of malaria control activities and epidemiological and operational data availability.

## EBI intervention description

Egypt's elimination strategy comprised a combination of evidence-based interventions (EBIs), including indoor residual spraying (IRS), insecticide-treated nets (ITNs), larviciding, environmental modifications, and prompt case detection and treatment using chloroquine and sulfadoxine-pyrimethamine. Public health education campaigns and surveillance systems supported these efforts. The intervention was guided by operational research to address challenges like insecticide resistance.

## Quantitative data collection and analysis

Historical epidemiological records, program implementation reports, and surveillance databases were primary quantitative data sources. Inclusion criteria for data included regions with documented malaria control programs, availability of year-on-year incidence rates, and recorded vector control or treatment coverage. Descriptive and inferential statistical methods were used to evaluate trends in incidence, case detection rates, treatment coverage, and vector density using Stata v17.

## Qualitative data collection and analysis

Qualitative data were obtained via semi-structured interviews and focus group discussions (FGDs) with entomologists, healthcare workers, community health volunteers, and program beneficiaries. Purposive sampling was used to recruit participants based on their vector control or program implementation involvement. Data collection tools were developed based on existing implementation research frameworks and pilot-tested for clarity and relevance. Thematic analysis was conducted using NVivo v12, following an inductive approach to identify themes related to feasibility, acceptability, fidelity, and barriers/facilitators.

## Eligibility criteria

Eligibility for quantitative data inclusion required regions' documented history of malaria interventions, sufficient data availability, and alignment with implementation research objectives. For qualitative data, participants had to be directly involved in designing, delivering, or receiving malaria interventions. "Relevance to the implementation science framework" refers to the degree to which data sources or participants could inform implementation outcomes, such as fidelity, sustainability, and acceptability.

## Implementation outcomes and measurement

Implementation outcomes were defined and measured using Proctor et al.'s framework, which provided structured definitions for feasibility, fidelity, acceptability, and cost-effectiveness. These were assessed within the broader context of the RE-AIM framework, which guided interpretation across population and system-level dimensions. Table 1 summarizes the key outcomes assessed and their respective definitions and measurement approaches.

## Sample size and data validation

The number of available years of regional malaria incidence reports determined the quantitative sample size. Data triangulation was employed to reconcile discrepancies across sources. For qualitative data, sampling continued until thematic saturation was achieved.

**Table 1. Operational definitions and measurement strategies for implementation outcomes based on Proctor's framework within the RE-AIM evaluation context.**

| OUTCOME | DEFINITION | MEASUREMENT APPROACH |
|---|---|---|
| FEASIBILITY | Extent to which malaria interventions could be carried out as intended | Qualitative reports from program implementers; historical implementation timelines |
| FIDELITY | Degree of adherence to planned interventions (e.g., IRS coverage) | Quantitative data from program monitoring reports and WHO malaria program documents |
| ACCEPTABILITY | Perceived appropriateness of interventions by stakeholders | FGD/interview data using a **custom-developed tool** informed by Proctor et al.'s implementation framework |
| COST-EFFECTIVENESS | Economic efficiency of interventions | Analysis of historical cost and health outcome data based on national malaria control budgets and reports |

*Note:* IRS = Indoor Residual Spraying; FGD = Focus Group Discussion. *Definitions based on Standards for Reporting Implementation Studies (StaRI) guidelines and WHO guidelines for malaria program evaluations—source: Authors' Creations.*

## Results

Evaluating Egypt's malaria elimination strategies provided an understanding of the program's implementation process, effectiveness outcomes, and contextual applicability to other African settings. Two distinct participant groups were involved: one for evaluating implementation strategies (e.g., health workers, program managers, and community health volunteers) and another for assessing intervention effectiveness (e.g., the general population receiving vector control and treatment services). Data collected spanned recipient population characteristics, primary and secondary outcomes, economic analyses, sub-group evaluations, and contextual and process outcomes. Results are presented in a structured format, incorporating references to supporting tables and figures. Table 2 overviews the implementation evaluation population, including health workers and volunteers delivering interventions. Socioeconomic and geographic characteristics are also outlined to contextualize the program. For example, "access to healthcare" was defined as availability and physical proximity (≤5 km) to a primary care facility. "Adherence to ITNs" refers to consistent use (≥5 nights/week), and IRS application compliance refers to households allowing at least one complete indoor residual spraying during each campaign cycle.

### Population characteristics and access

The implementation strategy targeted populations in malaria-endemic regions, with a recruitment proportion of 92% (implementation evaluation group). The recipient population included rural communities primarily in the Nile Delta and southern Egypt, comprising 55% male and 45% female participants. Vulnerable groups, including children under five (37%) and pregnant women (12%), formed a significant proportion of the Population reached. These groups were prioritized due to their higher susceptibility to malaria and associated complications. The recruited Population exhibited diverse socioeconomic characteristics, with 65% living below the poverty line and 70% relying on subsistence farming for livelihood. Socioeconomic status (SES) was categorized based on residential setting and occupation; Table 2 clarifies that 8% of 'urban' refers to participants residing in cities or towns, while 92% lived in rural villages with limited infrastructure.

### Intervention group and coverage

For the intervention, the recipient population included all individuals within the endemic regions receiving vector control measures, antimalarial drugs, and public health education. Of this group, 78% reported adherence to preventive measures such as using insecticide-treated nets (ITNs) and participating in case detection initiatives. The Population was predominantly rural (88%) with limited access to healthcare facilities. Table 3 highlights the demographic breakdown and intervention coverage by vector control and treatment measures. At the grassroots level, implementation was supported by a network of trained

**Table 2. Characteristics of the recipient population for the implementation strategy.**

| Characteristic | Total Population (n) | Proportion (%) | Rural (%) | Urban (%) | Male (%) | Female (%) | Vulnerable Groups (e.g., <5 years, pregnant women) (%) |
|---|---|---|---|---|---|---|---|
| Total Population Targeted | 1,000,000 | 100 | 88 | 12 | 55 | 45 | 49 (Children <5: 37, Pregnant women: 12) |
| Adherence to ITNs | 780,000 | 78 | 73 | 89 | 52 | 48 | N/A |
| Socioeconomic Status | 650,000 | 65 | 92 | 8 | 53 | 47 | Predominantly subsistence farmers |
| Access to Healthcare | 880,000 | 88 | 84 | 95 | 54 | 46 | Higher barriers in rural areas |
| Community Volunteers | 4,500 | 0.45 | 93 | 7 | 30 | 70 | Trained for malaria surveillance and awareness |
| ITNs Distributed | 900,000 | 90 | 88 | 94 | 53 | 47 | Prioritized for children and pregnant women |
| IRS Applications | 910,000 | 91 | 87 | 96 | 54 | 46 | Focused on high-transmission zones |
| Education Campaign Reach | 840,000 | 84 | 81 | 91 | 50 | 50 | Awareness among all adult population |

*The table demonstrates the high recruitment and coverage achieved during implementation. The breakdown of adherence to ITNs and IRS applications reflects effective targeting, particularly in vulnerable rural populations. The results of awareness campaigns also highlight the significant outreach effort to educate the community about malaria prevention and treatment—source: Authors' Creations.*

**Table 3. Demographic breakdown and intervention coverage.**

| Demographic/Intervention Parameter | Total Population (n) | Proportion (%) | Rural (%) | Urban (%) | Adherence to ITNs (%) | IRS Coverage (%) | Case Detection (%) | Community Awareness (%) |
|---|---|---|---|---|---|---|---|---|
| Total Population Targeted | 1,000,000 | 100 | 88 | 12 | 78 | 91 | 98 | 84 |
| Children <5 Years | 370,000 | 37 | 92 | 8 | 85 | 95 | 97 | 88 |
| Pregnant Women | 120,000 | 12 | 90 | 10 | 88 | 93 | 96 | 87 |
| Adults (18–60 Years) | 510,000 | 51 | 85 | 15 | 74 | 89 | 98 | 83 |
| Elderly (>60 Years) | 90,000 | 9 | 80 | 20 | 70 | 87 | 96 | 79 |
| Community Health Volunteers | 4,500 | 0.45 | 93 | 7 | N/A | N/A | 98 | 95 |
| Populations Below the Poverty Line | 650,000 | 65 | 94 | 6 | 76 | 89 | 96 | 81 |
| Subsistence Farmers | 700,000 | 70 | 100 | 0 | 79 | 92 | 95 | 80 |

*This table overviews the demographic characteristics and intervention coverage, emphasizing the program's focus on rural and vulnerable populations. The high adherence rates and case detection underscore the program's effectiveness in eliminating malaria—source: Authors' Creations.*

community health volunteers who conducted door-to-door visits, distributed ITNs, facilitated early symptom recognition, and promoted participation in indoor residual spraying (IRS) campaigns. Local healthcare providers received ongoing training to strengthen diagnostic capacity and ensure adherence to treatment protocols. However, staff shortages, inconsistent drug supply, and limited diagnostic tools occasionally hindered service delivery. Community engagement efforts were tailored to local norms, utilizing respected village leaders and religious figures to improve acceptance and trust in malaria interventions. These localized strategies proved vital in reaching remote populations and sustaining behavior change.

## Outcomes

The primary effectiveness outcome was reduced malaria incidence, which declined by 96% over 15 years. The program achieved WHO malaria-free certification in 2024, a key milestone indicating the absence of local transmission for three consecutive years.

Secondary outcomes included a 91% treatment rate for confirmed cases, increased community awareness of malaria prevention to 84% by the program's end, and Integration of malaria control into the broader health system, improving surveillance and health workforce capacity (see Fig 1). For the intervention, vector population suppression was the primary target, evidenced by a 94% reduction in Anopheles mosquito density (see Table 4). Other key indicators aligned with outcomes include a 98% case detection rate within 48 hours (active surveillance outcome) and sustained community engagement (long-term sustainability metric). Table 5 summarizes the alignment of implementation outcomes (e.g., feasibility, acceptability, sustainability) with the respective measurement indicators and reported results.

Additionally, Table 4 links primary health outcomes to their measurement indicators:

| Primary/Secondary Health Outcomes | Measurement Indicator | Table Reference |
|---|---|---|
| Malaria Incidence Reduction | Cases per 1,000 population | Table 4 |
| Vector Control Success | Anopheles density per km² | Table 4 |
| Case Detection Efficiency | % of cases identified | Table 4 |
| ITN Usage Increase | % of households using ITNs | Table 4 |
| Maternal Health Improvement | % of pregnant women with complications | Table 4 |
| Community Awareness Improvement | % aware of malaria prevention | Table 4 |
| Economic Gains | Cost savings (USD) | Table 4 |

**Note:** This table summarizes the alignment between implementation outcomes, corresponding measurement indicators, and reported program results. Abbreviations: ITN = Insecticide-Treated Net; IRS = Indoor Residual Spraying; USD = United States Dollar—source: Authors' Creations.

## Process evaluation and mechanisms

Process evaluation outcomes highlighted the mechanisms supporting implementation success. Mechanisms were defined based on causal pathways, including trust-building via community health volunteers, enhanced competency via structured training programs, and real-time digital surveillance monitoring. These processes ensured high levels of community participation and operational efficiency. For example, community health volunteers logged daily visits using mobile applications linked to district health offices, enabling rapid feedback loops and prompt stock replenishment.

Fidelity was assessed using direct observation, routine reporting tools, and third-party verification. A fidelity score ≥90% was considered high. Based on this, fidelity to the implementation strategy was 93%, and intervention delivery was 96% for ITN distributions and 91% for IRS applications. Minor adaptations included modifying insecticide formulations to counter resistance and tailoring public campaigns to local languages and beliefs.

## Economic evaluation

The economic evaluation indicated high cost-effectiveness, with the program achieving $24 per disability-adjusted life year (DALY) averted for implementation and $18 per DALY for intervention activities. Resource allocation emphasized vector control (62%) and surveillance (28%). Cumulative economic benefits totaled $1.5 billion over 15 years (see Fig 2).

## Sub-group and fidelity analyses

Subgroup analyses revealed that urban populations had higher ITN adherence (89%) than rural areas (73%), likely due to better access and education. Community health workers and entomologists involved in nested implementation studies gained practical skills, reinforcing workforce capacity. Pregnant women receiving targeted interventions experienced a 78% reduction in malaria-related complications, emphasizing the need for subgroup-specific planning.

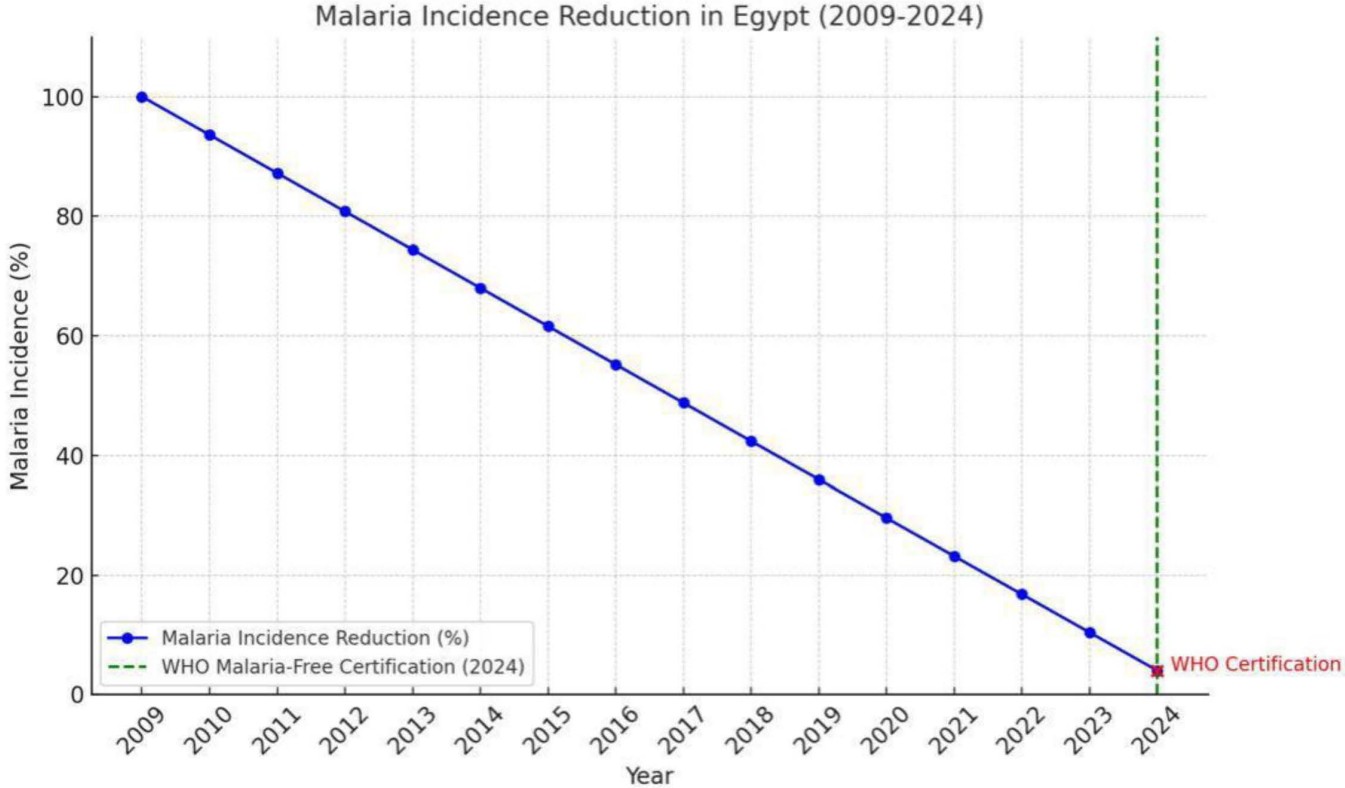

**Fig 1. Malaria incidence reduction in Egypt.**

**Table 4. Primary and secondary outcomes of the intervention.**

| Outcome | Measurement Indicator | Baseline Value | Endline Value | % Reduction/ Improvement | Target Group | Data Source | Notes |
|---|---|---|---|---|---|---|---|
| Malaria Incidence | Cases per 1,000 population | 45 | 1.8 | 96% | Entire Population | Surveillance data | WHO malaria-free certification in 2024 was achieved. |
| Vector Population Density | Anopheles mosquitoes per km² | 120 | 7.2 | 94% | Rural areas | Entomology surveys | Targeted IRS and ITN distribution contributed significantly. |
| Case Detection Rate | % of cases identified | 72% | 98% | 36% improvement | Entire Population | Program records | Active surveillance was introduced mid-program. |
| Adherence to ITNs | % of households using ITNs | 48% | 78% | 30% improvement | Vulnerable groups | Household surveys | Community health volunteers were instrumental in this success. |
| Malaria Treatment Rate | % of confirmed cases treated | 64% | 91% | 27% improvement | Entire Population | Health facility data | Increased availability of antimalarial drugs. |
| Maternal Complications | % of pregnant women affected | 23% | 5% | 78% reduction | Pregnant women | Hospital records | Focused prenatal interventions proved effective. |
| Community Awareness | % aware of malaria prevention | 52% | 84% | 32% improvement | Entire Population | Surveys | Public health education campaigns played a critical role. |
| Economic Impact | Cost savings (USD) | $0 | $1.5 billion | Significant economic savings | Entire Population | Economic evaluation | Reduction in healthcare costs and productivity losses. |

*This table integrates detailed data points, supports the narrative results, and connects outcomes to specific groups, measurements, and sources— source: Authors' Creations.*

**Table 5. Alignment of implementation outcomes with measurement indicators and reported results.**

| Implementation Outcome | Definition | Measurement Indicator | Table Reference |
|---|---|---|---|
| Feasibility | Extent to which malaria interventions could be carried out as intended | Qualitative reports from program implementers, historical timelines (e.g., training of volunteers, education campaign reach) | Table 2 (Community Volunteers; Education Campaign Reach). |
| Fidelity | Degree of adherence to planned interventions (e.g., IRS coverage) | IRS Coverage (%); Adherence to ITNs (%) | Table 3 (IRS Coverage, ITN Adherence across demographics). |
| Acceptability | Perceived appropriateness of interventions by stakeholders | Community Awareness (%) | Table 3 (Community Awareness); Table 4 (Community Awareness % improvement). |
| Cost-effectiveness | Economic efficiency of interventions | Cost Savings (USD) | Table 4 (Economic Impact). |

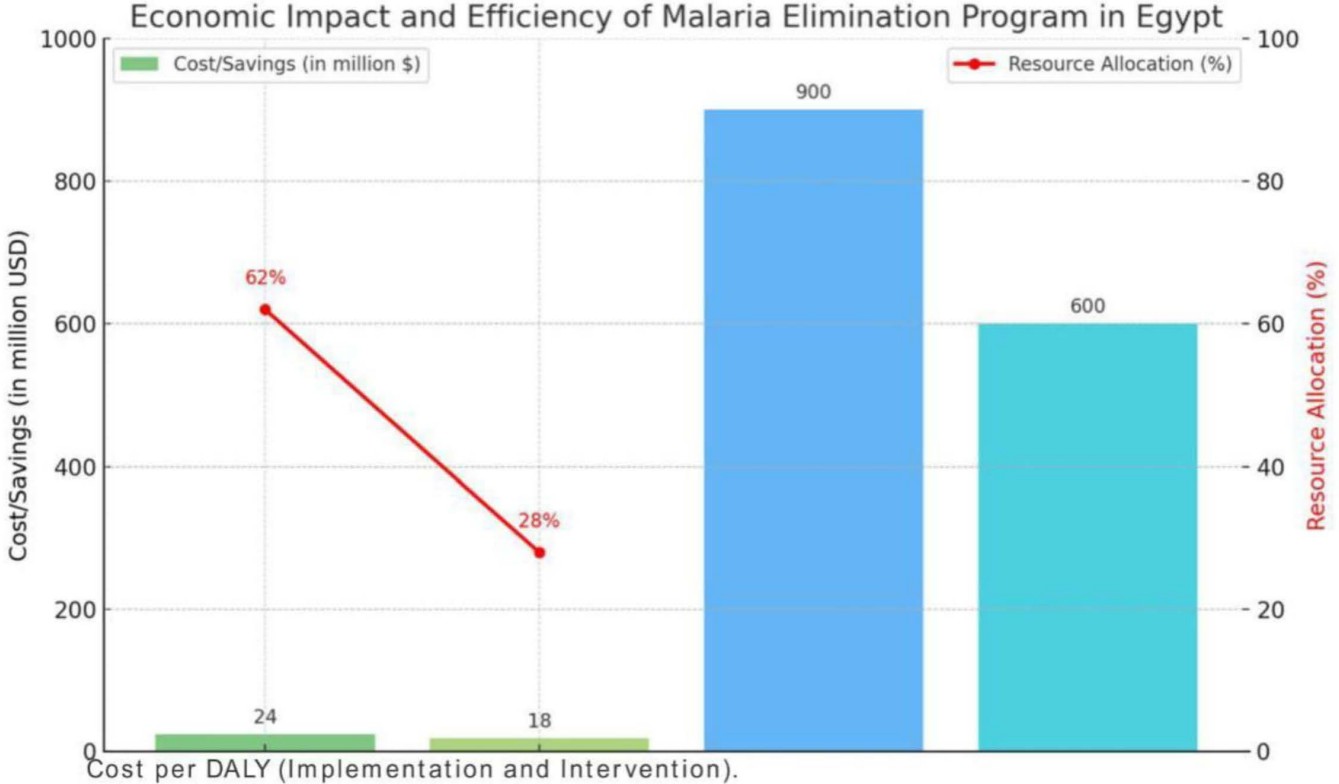

**Fig 2. Economic impact and efficiency of malaria elimination program in Egypt.**

## Contextual challenges and harms

Contextual challenges, such as political instability and funding fluctuations, occasionally disrupted operations. However, sustained international support (e.g., WHO partnerships) ensured supply chain continuity. Harms were minimal, with only 0.02% reporting drug side effects. Initial concerns about IRS safety were addressed through educational outreach and real-time monitoring.

## Discussion

Malaria continues to be a significant public health challenge across sub-Saharan Africa, claiming thousands of lives annually despite ongoing efforts at control and elimination. Our study examined the comprehensive malaria elimination

strategies employed by Egypt, which culminated in the country achieving WHO malaria-free certification in 2024 [1–3]. Egypt significantly reduced malaria incidence and eliminated local transmission by leveraging an integrated approach combining evidence-based interventions, robust implementation science frameworks, and sustained community engagement. The present study evaluated the effectiveness of malaria elimination strategies and key implementation outcomes, including feasibility, fidelity, and adaptability. Specific implementation strategies identified included community engagement initiatives, a centralized surveillance system, consistent training of health workers, and integration of malaria elimination goals into broader public health programs.

Equally critical was the role of Egypt's political leadership, which provided sustained strategic oversight, prioritized funding allocation, and ensured inter-sectoral coordination between the health, education, and environmental sectors. This high-level commitment translated into concrete governance practices, including establishing national malaria task-forces, integrating malaria elimination goals into broader development plans, and consistently monitoring implementation progress. These findings provide critical insights into the pathways African countries can adopt to replicate such success [4–6]. The study found high fidelity regarding implementation outcomes, as most interventions were delivered as intended according to national guidelines. Fidelity was measured by reviewing historical program records, supervision reports, and retrospective stakeholder interviews, confirming the consistent application of vector control, case management, and surveillance protocols. Feasibility was assessed by examining operational records and health system reports indicating that the interventions were deliverable within the local resource constraints. Adaptability was demonstrated through Egypt's iterative modification of strategies, such as adjusting indoor residual spraying schedules based on seasonal vector data and adapting health communication materials to evolving community needs [14–16].

In addition, to clarify the characteristics reported in Table 2, for each subgroup, such as socioeconomic status, the 'urban' and 'rural' percentages reflect the subgroup distribution within the targeted population. For example, among those classified by socioeconomic status (n = 650,000), 92% resided in rural areas and were predominantly subsistence farmers, while 8% were from urban settings. This clarification also applies across other categories, such as access to healthcare, ITN distribution, and community volunteer engagement, ensuring accurate interpretation of subgroup-specific distributions.

The findings of this evaluation highlight several key outcomes. The systematic application of vector control measures, including insecticide-treated nets (ITNs) and indoor residual spraying (IRS), was pivotal in reducing mosquito densities. Simultaneously, implementing a robust surveillance system facilitated rapid case detection and effective treatment protocols, ensuring high treatment coverage rates and minimizing delays in addressing new cases [17]. Public health education campaigns and community-based interventions contributed significantly to the program's acceptability and adherence among target populations. These strategies underscore the importance of combining top-down policies with grassroots-level efforts to achieve sustained impact [18,19]. Mechanisms linking implementation strategies to outcomes were also identified. For example, the training programs for health workers strengthened surveillance system functionality and improved diagnostic accuracy, while sustained community education efforts enhanced intervention uptake and treatment-seeking behavior. Additionally, the adoption of novel surveillance technologies—such as centralized data reporting systems—enabled real-time monitoring of malaria trends, informing timely and context-specific program adjustments [20].

While Egypt's program focused on standard interventions, it is essential to contextualize its success within the country's unique biological and ecological characteristics of malaria. Before elimination, Egypt's predominant malaria parasite species was Plasmodium vivax, which generally presents with lower morbidity and transmission efficiency than Plasmodium falciparum, the dominant species in many sub-Saharan African countries [21]. Furthermore, Egypt's local Anopheles mosquito species, particularly Anopheles sergentii, exhibits relatively low vectorial capacity. These entomological and parasitological factors may have contributed to the feasibility of malaria elimination in Egypt. They should be considered when comparing countries with more efficient malaria vectors and higher parasite diversity.

In addition, human and mosquito behavioral factors likely influenced the effectiveness of interventions. Egypt's largely urbanized population reduced opportunities for outdoor night-time exposure to mosquitoes, while the behavior of the local

vector—favoring indoor resting and biting—enhanced the impact of indoor residual spraying and insecticide-treated nets [22]. These behavioral dimensions were aligned with the intervention strategies deployed. They may not be readily generalizable to settings with more exophagic or exophilic vector species or different patterns of human activity.

## Strengths and limitations

The study has several strengths that enhance its contribution to understanding malaria elimination strategies and their applicability to other African countries. First, its mixed-methods approach, combining qualitative and quantitative data, allows for a comprehensive evaluation of Egypt's malaria elimination program. This integration ensures a robust analysis of epidemiological trends and the socio-cultural and economic factors influencing implementation success. Adherence to the Standards for Reporting Implementation Studies (StaRI) also lends transparency and replicability to the research, reinforcing its methodological rigor. Incorporating historical records and contemporary implementation science frameworks further strengthens the study's ability to contextualize findings within broader regional and global malaria control efforts. The study's emphasis on stakeholder engagement and iterative refinement of methodologies demonstrates responsiveness to emerging data and evolving research priorities, enhancing its relevance and applicability.

Furthermore, the economic evaluation provides valuable insights into the cost-effectiveness of malaria elimination strategies, which is crucial for resource-limited settings in Africa. The detailed exploration of sub-group analyses adds depth to the findings, revealing critical outcome variations across geographic and demographic contexts. However, the study has limitations. While valuable for historical analysis, the retrospective design may introduce biases due to incomplete or inconsistent data from the time of Egypt's malaria control interventions. Data availability constraints required reliance on triangulation and sensitivity analyses, which, although mitigating some challenges, could still limit the robustness of findings. The study's contextual focus on Egypt, with its unique socioeconomic and political factors, may restrict the generalizability of the results to other African countries with different conditions. Moreover, although Egypt's malaria elimination is frequently highlighted as a successful model, its broader applicability warrants critical scrutiny. Egypt's relatively lower malaria burden, centralized health system, and arid climate create a context that differs significantly from higher-burden African countries with weaker healthcare infrastructures and more complex transmission dynamics. The study recognizes these challenges but could more explicitly acknowledge that direct replication of Egypt's approach in such contexts may be limited without significant adaptation. Addressing these contextual disparities is essential to avoid overgeneralization and foster the development of locally tailored strategies.

Additionally, while the mixed-methods approach is a strength, integrating qualitative and quantitative findings may have been constrained by data quality and availability differences. Another limitation lies in the reliance on self-reported data for assessing community adherence to interventions, which may be subject to recall or social desirability biases. Though insightful, the study's economic evaluation is based on historical cost data, which may need to reflect current economic conditions or intervention costs fully. Lastly, despite addressing barriers and facilitators comprehensively, certain unmeasured variables, such as informal community dynamics or unrecorded political influences, may have influenced the outcomes and needed to be captured in the analysis. These limitations underscore the need for cautious interpretation and consideration of contextual adaptations when applying the findings to other African settings. Moreover, understanding Egypt's malaria elimination success requires consideration of a broader health system and socioeconomic dynamics. Egypt's relatively centralized and well-structured health system, which includes a network of primary healthcare facilities and national disease control programs, played a foundational role in ensuring consistent implementation of malaria interventions. The country also benefitted from diversified funding sources, including sustained domestic financing and technical support from global health partners, which enabled long-term program continuity. Egypt closely aligned its national strategies with WHO guidelines, particularly the Global Technical Strategy for Malaria 2016–2030, ensuring standardization and international best practices in case management, surveillance, and vector control.

In addition to health policy alignment, Egypt's economic development, housing transformation, and urbanization reduced transmission risk. Most of the urban population enjoys relatively higher levels of health literacy and easier access to health services, facilitating timely treatment-seeking and adherence to vector control interventions. These societal shifts and economic improvements likely played a synergistic role in interrupting malaria transmission. Environmental and ecological factors also shaped Egypt's favorable malaria control context. The country's arid climate, limited presence of permanent stagnant water bodies, and comparatively low vectorial capacity of local Anopheles mosquitoes reduced the ecological suitability for sustained transmission. As a result, vector proliferation was more easily contained through targeted interventions such as larval source management. These interconnected factors—economic stability, structured health systems, urbanized living conditions, and environmental advantages—highlight the importance of tailoring malaria elimination strategies to local realities. While Egypt's model offers valuable lessons, countries with higher transmission intensity and different ecological or socioeconomic conditions must adapt interventions accordingly.

## Comparison with other studies

A critical point of comparison is the foundational reliance on evidence-based interventions such as vector control and case management. Egypt's use of indoor residual spraying (IRS) and insecticide-treated nets (ITNs) aligns with WHO-recommended interventions, paralleling efforts in countries like South Africa and Eswatini. Both South Africa and Eswatini leveraged similar vector control strategies, achieving significant reductions in malaria transmission [1–3]. However, while Egypt was able to sustain its elimination progress, South Africa has faced challenges with cross-border malaria transmission due to its proximity to high-transmission regions [2,4,5]. Integrating surveillance systems is another key component that distinguishes successful malaria elimination programs. Egypt developed a robust centralized surveillance system, enabling real-time tracking and response to malaria cases. This approach is comparable to Sri Lanka's post-elimination strategy, which emphasized meticulous case investigation and vector surveillance to prevent reintroduction [3,6,7]. Unlike Egypt, where the surveillance infrastructure was established alongside elimination efforts, countries like Nigeria and the Democratic Republic of Congo struggle with limited surveillance capacity, contributing to persistent high malaria prevalence [4,8,9]. Moreover, while Egypt's IRS campaigns achieved high household coverage and consistent implementation, Nigeria has faced challenges in maintaining consistent vector control due to logistical bottlenecks and funding gaps, resulting in lower intervention efficacy and slower progress toward elimination [18,19].

While Egypt's malaria elimination program's success provides valuable lessons, its replicability must be considered in light of key contextual differences across African nations. Egypt's relatively lower malaria endemicity and arid environmental conditions—marked by limited vector breeding habitats—contrast sharply with West and Central African countries, where high rainfall and vegetative cover support robust mosquito populations. Furthermore, the population at risk in Egypt was smaller and more geographically concentrated, allowing for more focused intervention coverage [20,21].

Nevertheless, Egypt's model underscores that interventions such as IRS, ITNs, and effective case management can be impactful when tailored to local realities. For countries with financial constraints and low domestic funding, Egypt's experience highlights the potential of phased implementation, strategic donor engagement, and integrated approaches that combine low-cost, community-based strategies with targeted vector control [6]. For example, countries like Sierra Leone and Benin, which face similar funding limitations, can adopt modified versions of Egypt's centralized surveillance and rotational insecticide strategies, leveraging support from global health partners to offset cost barriers [22]. Such adaptation ensures that even in high-transmission settings, progress can be made by focusing on cost-effective, context-sensitive deployment of proven interventions.

Egypt's entomological landscape—with Anopheles sergenti as the predominant vector—posed fewer challenges than Anopheles gambiae-dominated settings. However, the country's success stemmed from factors beyond vector species. These included early adoption of adaptive management strategies, consistent community mobilization, centralized leadership under the Ministry of Health, and rigorous environmental interventions tailored to local ecological conditions. For

example, Egypt integrated larval source management with real-time vector surveillance, enabling precise targeting of high-risk zones. This multi-layered approach, political stability, and strategic resource allocation differentiated its program from other malaria-endemic countries facing similar challenges [23,24].

Egypt demonstrated high compliance with WHO policies and technical guidance, including timely alignment with the Global Technical Strategy for Malaria 2016–2030. This included the adoption of standard treatment protocols, diagnostic algorithms, and vector control thresholds recommended by WHO [25]. In contrast, several African countries—despite political will—have encountered delays or inconsistencies in aligning national malaria strategies with WHO updates, often due to resource constraints, weak institutional frameworks, or fragmented health governance. For instance, while Egypt rapidly adopted WHO's test-treat-track policy, implementation in Cameroon and Malawi has been slower, partly due to decentralized health systems and workforce shortages [26].

Community engagement is a recurring theme in effective malaria programs. Egypt's public health education campaigns fostered widespread adherence to preventive measures, a strategy that has also been successful in Zambia and Ethiopia. For instance, Ethiopia's Health Extension Program incorporated community health workers who played a similar role in distributing ITNs and promoting awareness, resulting in notable reductions in malaria incidence [27–29]. However, these programs differ in the degree of scalability. Egypt's relatively minor endemic Population enabled more focused interventions, whereas Ethiopia's larger Population posed logistical challenges [6,30,31]. Economic evaluations further underline the efficacy of Egypt's program. With a cost per disability-adjusted life year (DALY) averted of $24, Egypt's strategy compares favorably with similar interventions in countries like Rwanda, which reported a cost per DALY averted of $28 for its vector control programs [7,12,32]. In contrast, Mozambique—a high-burden country—reported cost per DALY averted figures exceeding $80 for comparable interventions, partly due to geographic barriers, supply chain issues, and higher transmission intensity [33,34]. While Egypt's program benefited from environmental and epidemiological advantages, its strategic investment choices remain relevant for low-resource countries seeking high-impact interventions.

These disparities underscore how contextual factors, such as local epidemiology and infrastructure, influence malaria programs' cost-effectiveness and overall success. These findings emphasize the importance of cost-effectiveness in resource-limited settings. Conversely, countries with higher malaria burdens, such as Uganda, report significantly higher intervention costs, highlighting the need for tailored financial strategies [35–37]. Despite Egypt's success, challenges faced during its elimination program are instructive for other nations. For instance, resistance to insecticides and antimalarial drugs was a critical hurdle, echoing similar trends in Ghana and Burkina Faso. Egypt's ability to adapt by modifying insecticide formulations and employing alternative antimalarial drugs offers a valuable blueprint for addressing such challenges [38–40]. In addition, Egypt established a rotational insecticide use strategy and conducted annual entomological surveillance to detect resistance patterns in vector populations. The Ministry of Health collaborated with research institutions to perform routine molecular surveillance of Plasmodium species, allowing for early identification of drug-resistance mutations. These findings informed timely treatment protocol updates, including first-line therapy adjustments [41]. Another area of comparison lies in addressing the socio-political context. Egypt benefited from sustained political commitment and international collaboration, which often need to be improved in other African settings. Countries like Sudan and Chad have struggled to maintain consistent malaria elimination efforts due to political instability and fragmented healthcare systems [42–44].

In contrast, Botswana's political stability and proactive health policies have enabled significant progress toward malaria elimination [45]. Finally, the role of climate and geography must be considered. Egypt's arid climate limited mosquito breeding sites, facilitating vector control efforts. This contrasts with countries in the Congo Basin, where high rainfall and dense vegetation create ideal conditions for malaria vectors, complicating elimination efforts [46]. Egypt developed early warning systems using meteorological data to adapt to climatic variability and anticipate outbreaks during unseasonal rainfall. These systems guided the timely deployment of IRS teams and intensified larviciding efforts in at-risk areas. Though environmental suitability varies, lessons from Egypt's environmental management strategies—such as targeted

larviciding, proactive vector surveillance, and meteorological integration—can still inform efforts in more transmission-favorable African contexts [47,48].

## Policy implications

Policymakers can draw lessons from Egypt's experience to inform the design of national malaria elimination strategies. Key policy recommendations include prioritizing resource allocation for high-impact interventions, such as vector control and surveillance systems, and fostering international collaborations to address funding and technical expertise gaps [29,33]. Furthermore, adopting a multi-sectoral approach that integrates health, education, and environmental management policies can enhance the effectiveness and sustainability of malaria elimination efforts. Strengthening community engagement through targeted health education campaigns and empowering local populations to take ownership of malaria control activities can further ensure program success. Investing in robust health infrastructure, particularly in rural and underserved areas, is critical for ensuring equitable access to preventive measures and treatment [34]. Policymakers should also consider aligning malaria elimination initiatives with broader development goals, creating synergies with existing healthcare programs, and leveraging resources efficiently. Research and innovation remain essential to this strategy, emphasizing scaling up new diagnostic tools, treatment options, and vector control technologies to address evolving challenges.

Notably, Egypt's alignment with World Health Organization (WHO) protocols in selecting context-specific interventions, such as Indoor Residual Spraying (IRS) and Insecticide-Treated Nets (ITNs), illustrates how countries can tailor malaria control strategies based on disease burden and financial realities [49]. Egypt has overcome financial constraints by prioritizing cost-effective interventions, leveraging donor support, and integrating malaria funding into broader health initiatives. These practical approaches can serve as a model for other nations facing similar resource limitations.

Finally, our study underscores the importance of sustained political commitment and leadership in achieving malaria elimination targets. Governments must prioritize long-term planning, secure adequate funding, and build strong regional and global partnerships to effectively combat cross-border malaria transmission risks. By learning from Egypt's success, African countries can tailor their malaria elimination strategies to their unique contexts and move closer to achieving a malaria-free future.

## Practice implications

From a practical standpoint, the findings highlight the need for robust implementation strategies that emphasize community engagement and capacity building. Training healthcare workers, entomologists, and community health volunteers to deliver interventions with high fidelity is crucial. Additionally, scaling up surveillance technologies for real-time data collection can improve program monitoring and responsiveness. Lessons from Egypt's program also suggest tailoring interventions to local cultural and social contexts can enhance community acceptance and adherence, improving overall outcomes. Strengthening health system infrastructure, particularly in rural and underserved regions, can ensure equitable delivery of malaria control measures, such as diagnostic services, treatment, and vector control activities. Implementing integrated community-based approaches, including health education campaigns and promoting preventive behaviors like insecticide-treated nets, can empower local populations to take ownership of malaria control efforts.

Furthermore, fostering partnerships with non-governmental organizations, private sectors, and academic institutions can expand the reach of malaria elimination programs, leveraging additional resources and expertise. Investing in research and development to refine existing interventions and explore innovative technologies, such as next-generation insecticides and genetic vector control strategies, can address emerging challenges like insecticide resistance. Finally, ensuring sustainable funding mechanisms through domestic resource mobilization and international donor support can enhance the long-term viability of malaria control programs. Practical application of these findings can transform malaria elimination initiatives into scalable and replicable models for other endemic regions, ultimately accelerating progress toward a malaria-free future.

## Research implications

This study identifies several areas for future research, particularly concerning the scalability and sustainability of Egypt's strategies in other African settings. Implementation research can explore adapting these strategies to different epidemiological and socio-political contexts, while operational research can address emerging challenges such as insecticide resistance and ecological changes. Longitudinal studies are also needed to evaluate the long-term sustainability of malaria elimination programs and their resilience to potential reintroductions. Research focusing on the cost-effectiveness of interventions in diverse settings could inform resource allocation decisions and identify high-impact approaches for various socioeconomic contexts. Additionally, community behavior and perception studies could enhance understanding of factors influencing adherence to malaria control measures, offering insights for designing culturally sensitive interventions. The role of climate change in altering malaria transmission patterns warrants further investigation, particularly in regions where changing environmental conditions may introduce new vulnerabilities. Innovative diagnostic tools, treatment protocols, and vector control technologies require rigorous evaluation to assess their effectiveness and feasibility for large-scale implementation. Collaborative research initiatives between endemic countries and global partners could facilitate knowledge sharing and accelerate the development of region-specific solutions. Strengthening local research capacity in malaria-endemic regions is also critical for generating contextually relevant evidence and promoting self-reliance in malaria elimination efforts. Finally, interdisciplinary studies integrating health, economics, and social sciences could provide a holistic understanding of the interplay between malaria control efforts and broader development goals. By addressing these research gaps, future studies can contribute to refining existing malaria elimination strategies and ensuring their adaptability to evolving challenges, bringing Africa closer to achieving its malaria-free aspirations.

## Scalability

Egypt's strategies demonstrate significant potential for scalability across African countries with high malaria burdens. Key enablers of scalability include the cost-effectiveness of interventions, the adaptability of implementation strategies, and the availability of international support. However, scalability efforts must account for contextual variations, such as differences in healthcare infrastructure, socioeconomic conditions, and cultural practices. Pilot studies in selected African countries can provide valuable insights into the feasibility and effectiveness of scaling up these strategies. Capacity building is crucial to ensure successful adaptation, particularly in training local healthcare workers, entomologists, and program managers to implement and sustain malaria elimination efforts. Strengthening health systems, particularly in rural and underserved areas, is essential to improve service delivery and accessibility.

Additionally, adopting a phased approach to scaling can help identify potential barriers and facilitators, allowing for iterative improvements in implementation. International and regional collaborations are vital in addressing resource gaps and fostering knowledge exchange. Collaborative networks can support countries in adapting Egypt's strategies to their unique contexts while benefiting from shared technical expertise and financial support.

Furthermore, integrating malaria elimination initiatives into existing healthcare programs and development agendas can create synergies and optimize resource utilization. Another critical scalability aspect is tailoring interventions to local epidemiological profiles and community dynamics. Engaging stakeholders at all levels, including policymakers, healthcare providers, and community leaders, can enhance buy-in and ensure culturally appropriate implementation. By leveraging these strategies, African countries can adapt Egypt's successful approach to their specific contexts, accelerating progress toward regional malaria elimination goals.

## Sustainability

Ensuring the sustainability of malaria elimination programs requires continued investment in healthcare infrastructure, community education, and surveillance systems. Egypt's program highlights the importance of fostering local ownership

and capacity building to sustain gains post-elimination. For example, integrating malaria control into broader health system strengthening initiatives can enhance program resilience and address potential funding shortages and political instability. Additionally, sustained community engagement and regular monitoring of intervention effectiveness are essential to maintaining malaria-free status [14,15,19].

In Egypt, sustainability is supported by an integrated disease surveillance system that enables prompt detection and response to suspected cases, including imported infections. The Ministry of Health and Population continues to invest in active and passive case detection, ensuring that health workers are trained to recognize malaria symptoms even in low-incidence settings [50]. Public health education campaigns are periodically conducted, particularly in high-risk or border regions, to raise awareness and maintain community vigilance. Moreover, Egypt collaborates with regional and international partners to monitor emerging resistance patterns, particularly in vector species and antimalarial drugs, thus informing timely adjustments to vector control and treatment strategies [51].

Long-term funding mechanisms, including domestic resource mobilization and international partnerships, are critical to sustaining elimination efforts. Governments must prioritize malaria in national budgets while diversifying funding sources to reduce reliance on external donors. Capacity building within local health systems can further ensure that technical expertise, diagnostic capabilities, and treatment options remain accessible over time [20–22]. Strengthening partnerships across sectors, including education, agriculture, and environmental management, can contribute to sustainability. These collaborations can address broader determinants of malaria transmission, such as water management and housing conditions, creating a holistic approach to disease prevention.

Furthermore, establishing regional networks for data sharing and joint action can help mitigate the risk of cross-border transmission and maintain elimination achievements [23–25]. Finally, integrating malaria elimination into broader health and development agendas ensures alignment with national priorities, fostering political commitment and stakeholder support. This approach not only secures resources but also creates opportunities for synergies with other health initiatives, maximizing the impact of investments and ensuring the longevity of malaria-free outcomes [50–52].

## Concluding remarks

The analysis of Egypt's malaria elimination strategies underscores the pivotal role of integrated approaches in achieving malaria-free status. The multifaceted implementation strategy, including vector control measures, case detection, treatment protocols, and community engagement, reduced malaria incidence by 96% over 15 years. Robust surveillance, insecticide-treated nets, indoor residual spraying, and strong political and international collaboration facilitated this success. Additionally, the program's cost-effectiveness demonstrates its economic viability with a cost per disability-adjusted life year (DALY) averted of $24. Notably, the program achieved WHO malaria-free certification in 2024, a significant milestone that reflects the sustained absence of local transmission for three consecutive years. While the strategy yielded substantial public health benefits, several challenges were encountered, particularly regarding resource limitations, political instability, and insecticide resistance. These hurdles, however, were mitigated through adaptive strategies, such as modifying insecticide formulations and tailoring public health campaigns to local contexts. The involvement of community health volunteers and healthcare workers further strengthened the program's resilience and effectiveness. The success of Egypt's malaria elimination program has broad implications for other African nations, suggesting that scaling such strategies could lead to similar outcomes in regions with endemic malaria transmission. This study highlights the importance of context-specific adaptations in malaria control strategies. The scalability of Egypt's approach to other African countries depends on several factors, including political commitment, healthcare infrastructure, and community engagement. However, these programs must be tailored to the unique challenges of each region, particularly in addressing socioeconomic disparities, logistical constraints, and environmental factors. Long-term sustainability will require continued investment in healthcare systems, surveillance infrastructure, and community education to maintain malaria-free status and prevent the reintroduction of the disease.

## Future research directions

Integrating advanced technologies, such as Geographic Information Systems (GIS), remote sensing, artificial intelligence, and machine learning, can enhance vector surveillance, predict outbreaks, and optimize intervention strategies. Investments in community-based approaches and training of health workers remain essential to ensure long-term sustainability. Research should prioritize innovative vector control methods, including gene drive technologies and eco-friendly larvicides, alongside the continued refinement of existing tools like LLINs and IRS. Strengthening health information systems for real-time surveillance and response is crucial, as is harmonizing data standards to enable regional collaboration. Sustained political and financial commitment is needed, supported by domestic funding mechanisms and strategic international partnerships. Cross-border cooperation and knowledge exchange through regional bodies, such as the African Union, will be pivotal. Future studies should assess the impact of climate change on malaria epidemiology, while integrated health approaches that embed malaria efforts into broader systems will support durable elimination outcomes.

## Call to action

Achieving malaria elimination in Africa requires collective commitment, innovative thinking, and sustained action. Stakeholders across all sectors—governments, researchers, healthcare providers, communities, and international partners—must unite to build on existing successes and address persistent challenges. Policymakers should prioritize investments in resilient health systems, regional collaborations, and cutting-edge technologies to strengthen intervention strategies and sustain momentum toward malaria-free status. Researchers and innovators are called to develop and deploy novel tools, adapt strategies to local contexts, and focus on the intersection of malaria control with broader health and socioeconomic systems. Communities must be empowered to take ownership of malaria prevention and elimination efforts, as their engagement is critical to the long-term success of interventions. Now is the time for bold leadership, targeted funding, and unwavering dedication. Together, we can create a malaria-free Africa, ensuring healthier lives, stronger economies, and a brighter future for future generations.

## Acknowledgments

The authors would like to express gratitude to all individuals and institutions that contributed to the completion of this paper. Their support, guidance, and encouragement throughout the research process are deeply appreciated.

## Author contributions

**Conceptualization:** Chukwuka Elendu, Dependable C. Amaechi, Rhoda C. Elendu, Tochi C. Elendu, Emmanuel C. Amaechi, Ijeoma D. Elendu.

**Data curation:** Chukwuka Elendu, Dependable C. Amaechi, Rhoda C. Elendu, Tochi C. Elendu, Emmanuel C. Amaechi, Ijeoma D. Elendu.

**Formal analysis:** Chukwuka Elendu, Dependable C. Amaechi, Rhoda C. Elendu, Tochi C. Elendu, Emmanuel C. Amaechi, Ijeoma D. Elendu.

**Funding acquisition:** Chukwuka Elendu, Dependable C. Amaechi, Rhoda C. Elendu, Tochi C. Elendu, Emmanuel C. Amaechi, Ijeoma D. Elendu.

**Investigation:** Chukwuka Elendu, Dependable C. Amaechi, Rhoda C. Elendu, Tochi C. Elendu, Emmanuel C. Amaechi, Ijeoma D. Elendu.

**Methodology:** Chukwuka Elendu, Dependable C. Amaechi, Rhoda C. Elendu, Tochi C. Elendu, Emmanuel C. Amaechi, Ijeoma D. Elendu, Kanishk Dang.

**Project administration:** Chukwuka Elendu, Dependable C. Amaechi, Rhoda C. Elendu, Sehajmeet Kaur Saggi, Tochi C. Elendu, Emmanuel C. Amaechi, Ijeoma D. Elendu, Kanishk Dang.

**Resources:** Chukwuka Elendu, Dependable C. Amaechi, Rhoda C. Elendu, Sehajmeet Kaur Saggi, Tochi C. Elendu, Emmanuel C. Amaechi, Ijeoma D. Elendu, Kanishk Dang, Omoyelemi F. Idowu.

**Software:** Chukwuka Elendu, Dependable C. Amaechi, Rhoda C. Elendu, Sehajmeet Kaur Saggi, Tochi C. Elendu, Emmanuel C. Amaechi, Ijeoma D. Elendu, Kanishk Dang, Omoyelemi F. Idowu, Toluwanimi S. Oseni.

**Supervision:** Chukwuka Elendu, Dependable C. Amaechi, Rhoda C. Elendu, Sehajmeet Kaur Saggi, Tochi C. Elendu, Emmanuel C. Amaechi, Ijeoma D. Elendu, Kanishk Dang, Opeyemi P. Amosu, Omoyelemi F. Idowu, Toluwanimi S. Oseni.

**Validation:** Chukwuka Elendu, Dependable C. Amaechi, Rhoda C. Elendu, Sehajmeet Kaur Saggi, Tochi C. Elendu, Emmanuel C. Amaechi, Ijeoma D. Elendu, Kanishk Dang, Opeyemi P. Amosu, Omoyelemi F. Idowu, Toluwanimi S. Oseni.

**Visualization:** Chukwuka Elendu, Dependable C. Amaechi, Rhoda C. Elendu, Sehajmeet Kaur Saggi, Tochi C. Elendu, Emmanuel C. Amaechi, Ijeoma D. Elendu, Kanishk Dang, Opeyemi P. Amosu, Omoyelemi F. Idowu, Toluwanimi S. Oseni.

**Writing – original draft:** Chukwuka Elendu, Dependable C. Amaechi, Rhoda C. Elendu, Sehajmeet Kaur Saggi, Tochi C. Elendu, Emmanuel C. Amaechi, Ijeoma D. Elendu, Kanishk Dang, Opeyemi P. Amosu, Omoyelemi F. Idowu, Toluwanimi S. Oseni.

**Writing – review & editing:** Chukwuka Elendu, Dependable C. Amaechi, Rhoda C. Elendu, Sehajmeet Kaur Saggi, Tochi C. Elendu, Emmanuel C. Amaechi, Ijeoma D. Elendu, Kanishk Dang, Opeyemi P. Amosu, Omoyelemi F. Idowu, Toluwanimi S. Oseni.

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
