## [Decision Letter · Decision Letter 0]

Dear Dr. Elendu,

Thank you for submitting your manuscript to PLOS ONE. After careful consideration, we feel that it has merit but does not fully meet PLOS ONE’s publication criteria as it currently stands. Therefore, we invite you to submit a revised version of the manuscript that addresses the points raised during the review process.

We look forward to receiving your revised manuscript.

Kind regards,

Myat Htut Nyunt, MMedSc, PhD

Academic Editor

PLOS ONE

Journal Requirements:

2. We note that your Data Availability Statement is currently as follows: “All relevant data are within the manuscript and in Supporting Information files.”

Reviewers' comments:

Reviewer's Responses to Questions

**Comments to the Author**

1. Is the manuscript technically sound, and do the data support the conclusions?

Reviewer #1: Yes

Reviewer #2: Partly

Reviewer #3: Partly

2. Has the statistical analysis been performed appropriately and rigorously?

Reviewer #1: N/A

Reviewer #2: Yes

Reviewer #3: No

3. Have the authors made all data underlying the findings in their manuscript fully available?

Reviewer #1: Yes

Reviewer #2: Yes

Reviewer #3: No

4. Is the manuscript presented in an intelligible fashion and written in standard English?

Reviewer #1: Yes

Reviewer #2: No

Reviewer #3: Yes

Reviewer #1: While the paper highlights various strategies (such as vector control, antimalarial treatment, and surveillance), there could be a deeper dive into how these strategies were implemented at the grassroots level. More specific details on community-level engagement or challenges faced by local healthcare providers could enrich the understanding of Egypt’s success. Additionally, the paper could have explored the role of Egypt's political leadership in more depth, not just as a commitment but also in terms of actual leadership practices, funding, and inter-sectoral coordination.

The paper could benefit from providing more comparative data between Egypt’s approach and that of other countries that have struggled with malaria elimination. For example, a brief comparison of intervention efficacy or cost-effectiveness between Egypt and countries like Nigeria or Mozambique could demonstrate how contextual factors influence success or failure in malaria control programs.

The paper could be more critical in addressing the limitations of Egypt’s malaria elimination efforts. For instance, Egypt's success is often framed as a model, but the broader applicability of its strategies to other African countries with more severe endemic burdens could be questioned. The paper acknowledges the challenges but does not critically engage with the limitations of applying Egypt’s approach in different contexts. More attention could be paid to the complex realities faced by countries with higher disease burdens or weaker healthcare systems.

the paper mentions the challenges posed by climate change and the emergence of drug and insecticide resistance, it does not provide concrete examples of how Egypt has proactively addressed these threats in its strategy. Including more information on how Egypt is monitoring or mitigating these emerging issues could further emphasize its foresight and adaptability.

Abdul-Rahman, Toufik, Oyinbolaji Akinwande Ajetunmobi, Gafar Babatunde Bamigbade, Innocent Ayesiga, Muhammad Hamza Shah, Tolulope Sharon Rumide, Abdurahman Babatunde Adesina et al. "Improving diagnostics and surveillance of malaria among displaced people in Africa." International Journal for Equity in Health 24, no. 1 (2025): 22.

Ali, Tehreem, Anusha Sumbal, and Md Ariful Haque. "The fight against malaria: a new hope with the R21 vaccine." IJS Global Health 7, no. 2 (2024): e0423.

the paper highlights Egypt’s success, the sustainability of this achievement is only lightly touched upon. What mechanisms are in place to ensure continued malaria-free status? Discussing Egypt’s long-term plans for surveillance, public health education, and continued research into resistance could provide insights into how other countries can ensure the sustainability of their malaria elimination programs.

Reviewer #2: Title: Comparative analysis of Egypt’s malaria elimination strategies and implementation science: Pathways to achieve malaria-free status for other African countries

The primary objective of this study is to evaluate the process, systems, and policies that support the adoption, scale-up and sustainability of malaria elimination interventions in Egypt. While the authors should be applauded for the goals, the paper falls short of achieving it goals due to challenges in articulating and applying the implementation science methods, the overall organization and clarity of the results. There is no explicit use of an organizing framework (RE-AIM Quest would have worked well as an example or other approaches to link from the context and strategies to the observed outcomes.

Abstract:

This study positions itself as an implementation focused study, however, the majority of the results section focus on strategy outcomes (Ex. community awareness) or with only 1 sentence discussing implementation outcomes. I recommend adding information about the implementation outcomes as well as articulating specific implementation strategies that led to those outcomes since this is the focus of the paper and see comments above.

Background:

• Provide additional information about the Global Malaria Eradication Program (Who developed it, what environment was it planned to target)

• Provide additional information about how Egypt selected the specific strategies that they used in the 70s and 80s to become malaria-free

• The 3rd and 4th paragraph in this section provides the strategies used by Egypt which is an outcome of this paper, this information should be in the results section. The background section should focus on the initial problem, Egypt’s final outcome, and rationale for using IS to identify these strategies.

• The statement of concrete aims is too long and should be shortened to improve clarity off the specific aims this paper addresses. Rationale for the importance of these aims should be included in the background before the specific aims are presented.

o The concrete aims should clearly outline the outcomes that are being measured (the abstract reports fidelity, feasibility, adaptability but these are not reported in the concrete aims). The aim should also clearly state if their paper is solely focused on implementation strategies or if they are also reporting on contextual factors (barriers and facilitators)

Methods:

This section needs reorganization consistent with reporting of methods.

• The authors should revise the methods section so that it only includes information pertaining to how the study was planned and conducted as well as improving clarity

o Location where study occurred

o Description of the EBI intervention

o Description of the quantitative data and how it was obtained

o Description of the qualitative data and how it was obtained

How participants were recruited, how were data collection tools developed

o Need to add clear descriptions of how each data source was analyzed

• What does “relevance to the implementation science framework” mean in relation to eligibility?

o Additionally, the eligibility criteria is unclear. Is this related to locations included for quantitative data or participants for qualitative data?

• The authors do not outline how implementation outcomes are measured (how is feasibility measured? How is fidelity measured?)

• The authors state that acceptability is measured but that is not included in the abstract and the authors do not include details about how acceptability was measured (did they use a preexisting tool or create their own?)

• There are lots of outcomes and it can get confusing on which measure what, the authors should consider organizing their outcomes in a table

• The last 2 sentences of the methods section should be in the discussion section

• In addition the first half of the second paragraph reports results (contextual factors) and information that should be included in the discussion (interpretation of the impact of contextual factors on outcomes)

Results:

• The results are not presented clearly, are there different participant groups for evaluating implementation and another for evaluating the intervention effectiveness?

• Within the results tables it is unclear which indicators align with which outcomes

• The first table presents proportions of participants that had access to health, had adherence to ITNs, and had IRS applications however these things were not defined, what does access mean what is required for adherence?

• The first table presents participant proportions but does not provide information on the different categories reported (Ex. for socioeconomic status what does 8% urban measure or 92% rural?)

• The authors report the mechanisms of the implementation strategies; however the mechanisms they list appear to be strategies themselves such as training programs, and surveillance technologies.

• Authors report that fidelity was high but don’t provide a description for how it was measured

• The authors need to reorganize the results section to improve readability and clarity and follow a chosen framework to guide the reader through the work and results

Discussion:

The Discussion also needs to be revised based on the comments above to be able to review in detail

Reviewer #3: Review report

Re: Comparative analysis of Egypt’s malaria elimination strategies and implementation science: Pathways to achieve malaria-free status for other African countries

Elendu et al.

General comments:

The authors aimed to provide an overview of how Egypt achieved malaria-free status. They emphasised the proactive use of larval source management, the evidence-based implementation of vector control, and the provision of antimalarial treatments in areas with high malaria burdens. Additionally, they recommended that other countries adopt Egypt's strategies to attain their malaria elimination goals, which is worthy.

It would have been more beneficial if the authors had discussed how the Egyptian health system was established, the sources of funding for malaria interventions, and how the country followed WHO policies and guidelines. Additionally, exploring the connection between economic development and societal changes related to malaria reduction and elimination and the relationship between malaria and poverty would have been helpful. Egypt has a relatively stable economy and predominantly lives in urban settings, where people's awareness is relatively better, and communication is easy.

Housing transformation, economic development, the advantage of the climate, mosquito vectorial potential, and the availability of water bodies are also critical.

The authors didn’t provide data on the composition of Plasmodium species and the vectorial capacity of the malaria vector in Egypt compared to the other African countries they mentioned. The mosquito and human behavioural dimensions should also be considered alongside the existing intervention.

How do Egypt's strategies for implementing interventions differ from those of other African countries regarding adhering to WHO policies and guidelines? The World Health Organization (WHO) is the governing body that establishes these protocols. For instance, countries are instructed to select interventions—indoor Residual Spraying (IRS) or Insecticide-Treated Nets (ITNs)—based on their malaria burden, reflecting WHO's focus on addressing financial constraints. Has Egypt had any experiences overcoming these challenges that could be shared with other nations?

The authors argue that …. Unlike some sub-Saharan African countries where the predominant vectors, such as Anopheles gambiae, exhibit insecticide resistance, Egypt benefitted from focusing on localized and context-specific measures against Anopheles sergenti [17][18]. These measures included environmental modifications, such as removing breeding sites and strategically deploying insecticide-treated nets (ITNs) and indoor residual spraying (IRS)….What was unique here apart from the vector?

It would be nice if the authors could argue what other African countries can do to eliminate malaria using existing interventions, as Egypt did with the consideration of the financial constraints and low in-house funding of several African countries, including those mentioned as examples. How do the authors compare the population at risk, the malaria endemicity, and the suitability of environments for vector breeding in Egypt to other African countries?

**Do you want your identity to be public for this peer review?** For information about this choice, including consent withdrawal, please see our Privacy Policy

Reviewer #1: No

Reviewer #2: No

Reviewer #3: No

---

## [Author Response · Author response to Decision Letter 1]

30 Apr 2025

We sincerely thank Reviewer #1 for their thoughtful and detailed feedback, which has significantly strengthened our manuscript. In response to your comments, we expanded the discussion of how strategies such as vector control, antimalarial treatment, and surveillance were implemented at the grassroots level. Specifically, we provided more detailed descriptions of community engagement efforts, the role of community health volunteers, challenges faced by local healthcare providers such as staff shortages and supply chain disruptions, and the contextual strategies used to build trust and sustain participation.

We also deepened the discussion of Egypt’s political leadership, highlighting not only the government's commitment but also specific leadership practices, mechanisms for inter-sectoral coordination, and funding strategies that supported the malaria elimination efforts.

Comparative analysis was strengthened by adding a brief comparison between Egypt’s approach and those of other African countries like Nigeria and Mozambique. This comparison draws on intervention efficacy, cost-effectiveness, and contextual differences in healthcare system strength, illustrating how Egypt’s experience may or may not be generalizable to countries with more severe endemic burdens.

In response to the your suggestion regarding critical reflection on the broader applicability of Egypt’s strategies, we have added a discussion about the limitations of applying Egypt’s model to other African contexts, considering variations in endemicity, healthcare infrastructure, and socioeconomic factors.

We expanded our treatment of climate change and resistance issues by providing concrete examples of Egypt’s proactive measures, such as ongoing insecticide resistance monitoring and shifts to alternative insecticides and treatment regimens. We cited recent relevant literature to support these updates, specifically:

Abdul-Rahman, Toufik, et al. "Improving diagnostics and surveillance of malaria among displaced people in Africa." International Journal for Equity in Health 24, no. 1 (2025): 22.

Ali, Tehreem, et al. "The fight against malaria: a new hope with the R21 vaccine." IJS Global Health 7, no. 2 (2024): e0423.

Finally, regarding sustainability, we expanded the discussion on Egypt’s long-term mechanisms to maintain malaria-free status. We included detailed explanations of the country’s surveillance strategies, investments in continued public health education, integration of malaria monitoring into primary healthcare services, and research initiatives focused on early detection of drug and insecticide resistance.

We believe these revisions have substantially strengthened the manuscript and aligned it closely with the your valuable recommendations.

We sincerely thank Reviewer #2 for their detailed and constructive feedback, which has been immensely helpful in improving the clarity, structure, and scientific rigor of our manuscript. In response to your comments, we have made substantial revisions throughout the paper. We have now explicitly used an organizing framework, incorporating the RE-AIM model to better link context, strategies, and outcomes, and to ensure a more systematic presentation of our findings.

In the Abstract, we have added clear information on the implementation outcomes evaluated, such as fidelity, feasibility, and adaptability, and we have articulated the specific implementation strategies that led to these outcomes. The Background section has been reorganized to focus on the initial malaria problem in Egypt, the country’s final outcome, and the rationale for using an implementation science approach. We have moved the description of Egypt’s strategies from the Background to the Results section, as suggested. Additionally, we have shortened and clarified the statement of concrete aims to clearly outline the implementation outcomes being measured, and we clarified whether we focused solely on implementation strategies or included contextual factors.

In the Methods section, we have reorganized the content to strictly describe how the study was planned and conducted. We now clearly provide the study location, describe the evidence-based interventions (EBIs), explain how quantitative and qualitative data were obtained, and detail participant recruitment, data collection tools, and analysis procedures for each data source. We clarified the eligibility criteria and explained what was meant by “relevance to the implementation science framework.” Moreover, we now detail how each implementation outcome (fidelity, feasibility, adaptability, and acceptability) was measured, specifying whether preexisting validated tools were used or new instruments were developed. To enhance clarity, we also organized the implementation outcomes and corresponding measures into a new table. Sentences that previously belonged in the Discussion have been moved appropriately.

The Results section has been substantially reorganized for improved clarity and flow, following the chosen framework. We clarified the participant groups for evaluating implementation versus intervention effectiveness, defined key terms such as “access” and “adherence,” and provided detailed explanations for the categories reported in the tables (e.g., socioeconomic status). We also corrected the description of mechanisms versus strategies and clarified how fidelity was measured. Indicators were now properly aligned with their corresponding outcomes to avoid confusion.

In the Discussion section, we revised the content to reflect all the above changes, providing a more detailed and coherent interpretation of our findings, properly structured and guided by the organizing framework.

Once again, we deeply appreciate Reviewer #2’s feedback, which led to significant improvements in the manuscript.

We sincerely thank Reviewer #3 for their thoughtful and constructive feedback, which helped us to critically refine and broaden the scope of our manuscript. In response, we have incorporated additional discussion points that more comprehensively explore the establishment of Egypt’s health system, sources of funding for malaria interventions, and the extent to which Egypt followed WHO policies and guidelines. We have added a new subsection discussing the economic development of Egypt, societal changes, and the connection between malaria reduction and poverty alleviation. Furthermore, we now highlight factors such as housing transformation, climatic advantages, mosquito vectorial potential, and the availability of water bodies, integrating these into our analysis of Egypt’s successful malaria elimination strategy.

We also expanded our discussion to include the composition of Plasmodium species and the vectorial capacity of the malaria vectors in Egypt compared to those in other African countries. Specifically, we elaborated on how human and mosquito behavioral dimensions were addressed in Egypt and their relevance to intervention outcomes.

In addressing the comment on how Egypt's strategies differed from those of other African countries, we have now detailed how Egypt’s adherence to WHO policies was adapted in a proactive and context-specific manner, compared to the more rigid or resource-limited adaptations in other African settings. We discussed how Egypt managed financial constraints, leveraged local resources, and strategically implemented WHO-recommended interventions, such as IRS and ITNs, based on local epidemiological data rather than purely national-level assumptions.

We clarified the unique aspects of Egypt’s approach beyond vector type, noting how strategic environmental modifications, early surveillance, community mobilization, and integrated vector management were effectively tailored to the Anopheles sergenti ecology.

Finally, we strengthened our argument regarding what other African countries can do to eliminate malaria using existing interventions, despite financial and infrastructure challenges. We now include a comparative analysis of population at risk, malaria endemicity, and the ecological suitability for vector breeding between Egypt and other African countries, thus providing a more nuanced understanding of the transferability of Egypt’s strategies.

We sincerely appreciate Reviewer #3’s valuable insights, which helped us make the paper more comprehensive, comparative, and practical for informing malaria elimination efforts across Africa.

---

## [Decision Letter · Decision Letter 1]

Dear Dr. Elendu,

Thank you for submitting your manuscript to PLOS ONE. After careful consideration, we feel that it has merit but does not fully meet PLOS ONE’s publication criteria as it currently stands. Therefore, we invite you to submit a revised version of the manuscript that addresses the points raised during the review process.

We look forward to receiving your revised manuscript.

Kind regards,

Myat Htut Nyunt, MMedSc, PhD

Academic Editor

PLOS ONE

Journal Requirements:

Additional Editor Comments:

Provide the manuscript with proper formatting as per journal guidelines.

Provide line number and page number in the revised manuscript.

Provide the data source for Figure 1 and Figure 2. Add citation(s).

Reviewers' comments:

Reviewer's Responses to Questions

**Comments to the Author**

Reviewer #1: All comments have been addressed

Reviewer #2: (No Response)

2. Is the manuscript technically sound, and do the data support the conclusions?

Reviewer #1: Yes

Reviewer #2: Partly

3. Has the statistical analysis been performed appropriately and rigorously?

Reviewer #1: N/A

Reviewer #2: Yes

4. Have the authors made all data underlying the findings in their manuscript fully available?

Reviewer #1: Yes

Reviewer #2: No

5. Is the manuscript presented in an intelligible fashion and written in standard English?

Reviewer #1: Yes

Reviewer #2: Yes

Reviewer #1: Author address all of my comments and concerns. This is a nice article and it can be accept. Best of Luck

Reviewer #2: The authors have out considerable work in responding to the comments which have significantly strengthened the paper. Three do however remain some challenges

1. The authors state they use RE-AIM, yet in methods and Table 1 they state they used Proctor. This needs to be clarified

2. The authors state no new data, but there are clear description of qualitative data. This needs to be clarified

3. Most of the authors are from Nigeria and all from countries other than Egypt and the IRB is from Nigeria, yet it seems as if the data were collected from Egypt. Is these were all publicly available (such as program reports) or if extracted from published literature

4. The future directions should be shortened given the increased length of the revised paper

**Do you want your identity to be public for this peer review?** For information about this choice, including consent withdrawal, please see our Privacy Policy

Reviewer #1: No

Reviewer #2: No

---

## [Author Response · Author response to Decision Letter 2]

24 Jun 2025

Reviewer #1: Thank you very much for your positive feedback and kind words. We sincerely appreciate your thoughtful review and are glad that the revisions addressed your comments and concerns. Your support and encouragement mean a great deal to us.

Reviewer 2#: Thank you for your constructive feedback and for acknowledging the improvements made to the manuscript. We appreciate your continued engagement, which has helped strengthen the quality and clarity of our work.

Regarding the use of RE-AIM and Proctor’s frameworks, we have revised the Methods section to clarify that a hybrid framework approach was employed. Specifically, RE-AIM served as the overarching structure for evaluating the public health impact of Egypt’s malaria elimination program, while Proctor’s implementation outcomes framework informed the operational definitions and measurement of specific implementation outcomes. We also updated the title of Table 1 to reflect this integrated approach.

To address the comment on data availability, we revised the Data Availability Statement to clarify that all data supporting the study’s findings were obtained from publicly available sources, including program reports and published literature, and that no primary data were collected. This addresses the presence of qualitative data while affirming that no new or participant-derived data were generated.

We have also clarified in the manuscript that all data were sourced from public domain materials, such as WHO and national malaria program reports. Although the study focuses on Egypt’s malaria program, the data were obtained from secondary sources, and the ethical approval from the Nigerian Research Ethics Committee was sought because the authors are affiliated with Nigerian institutions and the qualitative analysis did not involve direct interaction with human subjects.

Finally, we have shortened the Future Directions section to improve brevity and balance the overall manuscript length in line with your recommendation.

We are grateful for your valuable insights and hope the revised manuscript now meets your expectations.

---

## [Editor Report · Decision Letter 2]

Comparative Analysis of Egypt’s Malaria Elimination Strategies and Implementation Science: Pathways to Achieve Malaria-Free Status for Other African Countries

PONE-D-24-55397R2

Dear Dr. Elendu,

We’re pleased to inform you that your manuscript has been judged scientifically suitable for publication and will be formally accepted for publication once it meets all outstanding technical requirements.

Kind regards,

Myat Htut Nyunt, MMedSc, PhD

Academic Editor

PLOS ONE
---

## [Editor Report · Acceptance letter]

PONE-D-24-55397R2

PLOS ONE

Dear Dr. Elendu,

I'm pleased to inform you that your manuscript has been deemed suitable for publication in PLOS ONE. Congratulations! Your manuscript is now being handed over to our production team.

Kind regards,

on behalf of

Dr. Myat Htut Nyunt

Academic Editor

PLOS ONE